



# What can we learn from global disaster records about multi-hazards and their risk dynamics?

Wiebke S. Jäger[1], Marleen C. de Ruiter[1], Timothy Tiggeloven[1], Philip J. Ward[1,2]

[1]Institute for Environmental Studies (IVM), Vrije Universiteit Amsterdam, Amsterdam, The Netherlands
[2]Deltares, Delft, The Netherlands

*Correspondence to*: Wiebke S. Jäger (w.s.jaeger@vu.nl)

**Abstract.** Recent studies have been reporting more extreme, compounding impacts from multi-hazards than from single hazard events owing to complex interrelationships of hazard, exposure and vulnerability in a multi-hazard setting. However, our current understanding of multi-hazard impacts is primarily based on case studies of individual events. To complement
this, we examine the disaster records of the global emergency events database EM-DAT for the period 2000 – 2018 for evidence of multi-hazard risk dynamics. We develop an algorithm to identify multi-hazard events which uses the information on associated hazards as well as spatiotemporal relationships between disaster records in EM-DAT. We then perform a statistical analysis to assess potential risk dynamics in reported impacts of selected hazard pair types. We identified that twice as many hazards are part of multi-hazard events when considering a spatial overlap of at least 25% and a time lag of at
most 1 year between disaster records in addition to the information of associated hazards. These multi-hazard events account for 78% of the total damages, 83% of the total people affected and 69% of the total deaths in the reported disasters. The statistical comparison indicates that there are different patterns of how impacts compound depending on the impact metric as well as the hazard type. However, as a general trend, hazard pairs seem to have at least as or more impact than two isolated single hazards. To capture the patterns and to integrate them into risk analysis and decision making, we propose the
development of generic archetypes of multi-hazard risk dynamics. Despite the well-known limitations of EM-DAT related to completeness of the records as well as reliability of the impact data, which prevents detailed analyses of the data, we found the database useful for exploring high-level patterns at the global scale. Nonetheless, the uncertainties and limitations encountered highlight that future research should be directed at improving and supporting the multi-hazard and impact information in EM-DAT.

## 1 Introduction

Multiple studies have reported disproportionate impact amplifications during multi-hazard or compound events (e.g., see Gill and Malamud 2016; Zscheischler et al. 2018; de Ruiter et al. 2020). These amplifications can arise from several different elements in the multi-hazard context that interrelate with each other, leading to changes in impact (De Angeli et al. 2023). The interrelationships can be on the hazard, exposure as well as vulnerability level and it is widely recognized that
disregarding them can lead to an over- or underestimation of risk (Leonard et al. 2014; Zscheischler and Seneviratne 2017;





Hillier et al. 2020; de Ruiter and van Loon 2022; De Angeli et al. 2022; Ward et al. 2022) as well as ineffective or even harmful risk reduction strategies (de Ruiter et al. 2021; Ward et al. 2020; Hurk et al. 2023).

In this article, we follow the UNDRR definitions for risk, hazard, exposure and vulnerability (UNDRR 2017). Moreover, we use the term "multi-hazard impact" for impact generated from multiple hazards and accounting for all interrelationships on the hazard, exposure and vulnerability level following (Ward et al. 2022) and the term "multi-hazard risk dynamics" for changes in risks caused by those interrelationships. Table 1 provides an overview of the key risk definitions used in this article.

So far, hazard-hazard interrelationships have been researched most of the different types of interrelationships and several classification systems have been proposed (Gill and Malamud 2014; Liu, Siu, and Mitchell 2016; van Westen and Greiving 2017; Tilloy et al. 2019; Zscheischler et al. 2020; De Angeli et al. 2022). Though the terms differ across systems, they describe similar and overlapping concepts including statistical dependence between hazards, amplifications of hazard magnitude or triggering relationships. Methodological reviews and guidelines for quantifying the interrelationships have also been published (Tilloy et al. 2019; Bevacqua et al. 2021). Understanding and accounting for the hazard interactions is important, because they can lead to an impact that is different than the sum of the single-hazard effects (Kappes et al. 2012; Terzi et al. 2019).

Interactions on the exposure and vulnerability level have been less extensively researched, but examples of different types of changes in exposure and vulnerability been identified. For example, changes in exposure can arise, due to migration and evacuation (Tierolf et al. 2023) or due to losses and damages from a previous hazard that are not yet recovered (De Angeli et al. 2022). Furthermore, de Ruiter and van Loon (2022) discuss the complex interactions between hazards and vulnerability and identify key types of changes in vulnerability, such as the effects of an earlier hazard on the vulnerability at the time of a second hazard. It has also been identified that a combined load from multiple hazards can cause higher damage than the summed damages of the separate hazards (Zuccaro et al. 2008; Li et al. 2012).

The above-mentioned efforts have focussed on analysis of or methods for hazard-hazard interactions, hazard-exposure interactions or hazard-vulnerability interactions. For the built environment, de Angeli et al. (2022) propose a comprehensive modelling framework that integrates all three types of interactions for an assessment of multi-hazard impact. In this way, such a framework can enable the detection of overall changes in impact and risk due to the multi-hazard context through modelling. Nonetheless, our current understanding of multi-hazard impact in past events still is limited, with most evidence, as described above, being from case studies.



**Table 1 Definitions of terms used in this article**

| Term | Definition | Source |
|---|---|---|
| Risk | A combination of hazard, exposure and impact vulnerability as illustrated by the conceptual equation: Hazard x Exposure x Vulnerability | (UNDRR 2017) |
| Hazard | A process, phenomenon or human activity that may cause loss of life, injury or other health impacts, property damage, social and economic disruption, or environmental degradation. | (UNDRR 2017) |
| Exposure | The situation of people, infrastructure, housing, production capacities and other tangible human assets located in hazard-prone areas. | (UNDRR 2017) |
| Vulnerability | The conditions determined by physical, social, economic, and environmental factors or processes which increase the susceptibility of an individual, a community, assets, or systems to the impacts of hazards. | (UNDRR 2017) |
| Multi-hazard | The selection of multiple major hazards that the country faces and the specific contexts where specific hazards may occur over time simultaneously, cascadingly or cumulatively over time, and taking into account interrelated effects. | (UNDRR 2017) |
| Multi-hazard impact /risk | Impact / risk generated from multiple hazards and the interrelationships between these hazards and considering interrelationships on the vulnerability and exposure level. | (Ward et al. 2022) |
| Multi-hazard risk dynamics | Changes in risk or impact caused by interrelationships on the hazard, vulnerability or exposure level as compared to a case of no interrelationships. | This article |

The primary aim of this study is to explore the role of multi-hazard risk dynamics in globally reported disaster impacts to add to the existing body of knowledge based on case studies. To this end, we use the disaster records the emergency events data base EM-DAT (Delforge et al. 2023), which is, to our knowledge, the only publicly data available source for disaster

events including quantitative information on socio-economic impacts with global coverage. This database is widely-used in disaster risk science (Jones, Guha-Sapir, and Tubeuf 2022) and has been used before for multi-hazard analyses, in particular to classify historical disasters into different types of multi-hazard events by levering the information on main and associated hazards of each disaster record (Lee et al. 2024).

However, there are several challenges related to the use of EM-DAT for multi-hazard analyses, which apply to other global impact databases as well. First of all, EM-DAT has well-known issues related to reporting biases (Gall, Borden, and Cutter 2009) as well as the general reliability of the impact data (Guha-Sapir and Below 2002; Moriyama, Sasaki, and Ono 2018; Panwar and Sen 2020). In addition, the database records disasters from a single hazard perspective, though up to two associated hazards are included. An increasing trend in the reporting of associated hazards (Lee et al. 2024) as well as

recently developed global multi-hazard data sets (Claassen et al. 2023) also suggest that multi-hazards have been, and may still be, underreported with impacts being assigned only to a single main hazard. Moreover, hazards occurring simultaneously or in close succession at the same location have been reported in separate disaster records in multiple instances. An example of this is the Guatemala volcanic eruption and tropical cyclone that was described as detailed case

study in Gill and Malamud (2014). In light of these challenges and the fact that EM-DAT is widely used, a secondary aim of

this study is to examine the opportunities and limitations of this database for multi-hazard analysis.

To achieve these two aims, we set out to identify multi-hazard events in EM-DAT using not only the information on associated hazards, but also accounting for spatiotemporal relationships between disaster records following the common argumentation that hazards occurring close in time and space can cause significant risk dynamics (Kappes et al. 2012; de

Ruiter et al. 2020; De Angeli et al. 2022). As a consequence the identified events can consist of multiple disaster records. We then identify those impacts in the data set that have been caused by hazard pairs and those that have been caused by single hazards, and perform a statistical analysis to assess differences in impacts.

Statistical methods have previously shown to be useful for detecting differences in impacts. For example, Budimir, Atkinson,

and Lewis (2014) employed them to show that earthquake-landslide pairs result in more fatalities than earthquake single hazards. In this study, we compare both hazard pair impacts to single hazard impacts as well as hazard pair impacts to the combined impacts of two single hazards of the same type. The underlying idea is that the impacts of a hazard pair should equal the sum of impacts of two single hazards, if there are no multi-hazard risk dynamics. Conversely, a difference will point to multi-hazard risk dynamics.

**2 Data**

This study uses the international disaster database EM-DAT (Delforge et al. 2023) which contains information on natural hazards and their impacts together with the global data set of geocoded disaster locations GDIS (Rosvold and Buhaug 2021), which contains geospatial footprints of the impact areas.

**2.1 EM-DAT**

EM-DAT records events with substantial impact that are related to natural as well as technological hazards on country level from 1900 – present. Substantial impact is defined as an event which resulted in either at least ten deaths, at least 100 people affected, or a call for international assistance of an emergency declaration. Each entry corresponds to a disaster event on a country level. Events that span multiple countries are reported separately for each country.

Each disaster record in EM-DAT contains mandatory and optional fields. The mandatory fields relevant to this study are the unique event identifier, the country, the continent, the start year as well as the disaster type. We also use the optional fields, although data are frequently missing. Relevant optional fields are the disaster subtype, a first and second associated disaster, which represent subsequent or co-occurring hazards that may have contributed to the disaster impact, the start date and end date, as well as a number of human and economic impact variables.




EM-DAT uses a hierarchical classification system with types and subtypes for the main hazards, but not for the associated hazards. The types and subtypes are not clearly defined and the system for classifying the associated hazards is not documented. We use nine different hazard type terms throughout this article. Table 2 shows how we relate main hazards, based on the type and sub type, and associated hazards to these terms. EM-DAT records that contain other hazard types, either as main hazard or as associated hazard, are excluded for this analysis.

**Table 2 Hazard types used in this article versus terms used in EM-DAT**

| *Terms used in this article* | *Terms used in EM-DAT* | | |
|---|---|---|---|
| **Hazard types** | *Disaster type* | *Disaster subtype* | *Associated disaster* |
| Earthquake (eq) | Earthquake | Ground movement | Earthquake |
| Tsunami (ts) | | Tsunami | Tsunami/tidal wave, |
| Volcanic eruption (vo) | Volcanic activity | Ashfall, lahar, pyroclastic flow, lava flow | Volcanic activity |
| Landslide (ls) | Landslide | Landslide, rockfall, mudslide, avalanche (snow, debris, mudflow, rock) | Slide (land, mud, snow, rock), avalanche (snow, debris) |
| Coldwave (cw) | Extreme temperature | Cold wave | Cold wave |
| Heatwave (hw) | | Heat wave | Heat wave |
| Extreme wind (ew) | Storm | Convective storm, tropical cyclone, extra-tropical storm | Storm |
| Flood (fl) | Flood | Coastal flood, riverine flood, flash flood | Flood |
| Drought (dr) | Drought | All (Drought) | Drought |

In terms of impact, we consider the number of people affected, deaths, and damages. Throughout the following sections we will use the term "impact" to refer to these three quantities. Their definitions are:

- Number of People Affected: No. Injured, No. Affected and No. Homeless. No. Affected are the people needing immediate assistance due to the disaster. If only the number of families affected or houses damaged are reported, the figure is multiplied by the average family size for the affected area.
- Number of Deaths: confirmed fatalities directly imputed to the disaster plus missing people whose whereabouts since the disaster are unknown and so they are presumed dead based on official figures.
- Damages: refers to total economic damage in US $ adjusted for inflation.

EM-DAT is known to exhibit several biases due to having entire records missing rather than fields missing within records (Gall, Borden, and Cutter 2009). These include time bias, hazard-related bias, threshold bias, accounting bias, geographic bias as well as systemic bias. We exclude data from before the year 2000 to minimize time bias as recommended by the maintainers of EM-DAT (Delforge et al. 2023). However, other the other bias types remain, posing a limitation to this study. For example, heatwaves are known to be underreported in EM-DAT (Harrington and Otto 2020; Brimicombe et al. 2021).





Such biases only become apparent when comparing different disaster or hazard databases (Koç and Thieken 2018; Moriyama, Sasaki, and Ono 2018; Panwar and Sen 2020).

However, guidelines how to handle biases and missingness in disaster risk science are still lacking. Approaches for missingness differ across studies. Deletion, augmentation and imputation, or a combination of these, are most common for studies using EM-DAT as a primary or secondary data source (Jones, Kharb, and Tubeuf 2023). Deletion is simpler, but deemed inferior to augmentation and imputation because it poses a higher risk of introducing bias especially when data are not MAR (Nakagawa and Freckleton 2008). However, bias can be introduced by augmentation and imputation as well, if the data set used to develop those methods is already biased due to the missing cases.

We use two approaches for dealing with missing data. First, we use a deletion approach for distributions or statistics of hazard impacts. The approach, called 'available case analysis', utilizes only the observed data points for each variable. Because variables with few observations are less likely to be representative of the various possible underlying conditions in terms of hazard intensity, vulnerability and exposure than variables with many observations, we only conduct in-depth analyses for variables with at least 50 observations. Second, we use an imputation approach for total aggregate results that involve sums. Here, we assume missing values to be zeros. This is currently the standard approach in the literature though it inevitably leads to an underestimation of total impacts (Jones, Kharb, and Tubeuf 2023; Lee et al. 2024).

## 2.2 GDIS

GDIS is an open-source extension to EM-DAT and provides geographical approximations for main geophysical, meteorological, hydrological and climatological disaster types from 1960 – 2018 period (Rosvold and Buhaug, 2021). It includes spatial geometries for floods, storms, earthquakes, volcanic activity, extreme temperatures, landslides, and droughts, however not for wildfires. Overall, GDIS provides impact zones for almost 90% of these types of records.

The spatial geometries in GDIS correspond to administrative areas, as contained in the Global Administrative Areas database (GADM, n.d.). The geometries are derived from EM-DAT's country field or optional "Location", which lists the name(s) of the affected administrative area(s), or "Latitude" and "Longitude" fields, which provide coordinates for the location. Most locations can be described on the spatial resolution of administrative level 1 (typical state/province/region). The highest resolution corresponds to level 3 (district/commune/village) and the lowest resolution corresponds to level 0 (country). However, as hazards are unlikely to affect the entire area of an administrative region, the spatial geometries have to be regarded as crude approximates of the impact zones.




## 3 Method

Our method has two main parts and is outlined in Figure 1: firstly, developing multi-hazard data sets based on EM-DAT and GDIS; and secondly performing the statistical analyses of the impacts of single and multi-hazard.

The first part (developing multi-hazard data sets) has three steps. First, we create a georeferenced data set of disaster records by joining EM-DAT and GDIS. Second, we develop a data set of spatiotemporally overlapping disaster record pairs by assessing whether disaster records in EM-DAT have spatiotemporal overlaps. Third, we develop disaster record chains, hazard chains, and (multi-)hazards events, by using the spatiotemporally overlapping pairs together with information from the associated disaster fields in EM-DAT. Each of these steps is described in more detail in Sections 3.1.1 - 3.1.3.

The second part (statistical analysis of impacts) has two steps. First, we identify impacts of single hazards and of hazard pairs from the events for further analysis. Second, we compare the impacts of single hazards with impacts of hazard pairs. For simplicity, we restrict ourselves to hazard pairs as multi-hazards. Third, we compare the combined impacts of two individual single hazards with the impacts of hazard pairs. Each of these steps is described in more detail in Sections 3.2.1 - 3.2.3.

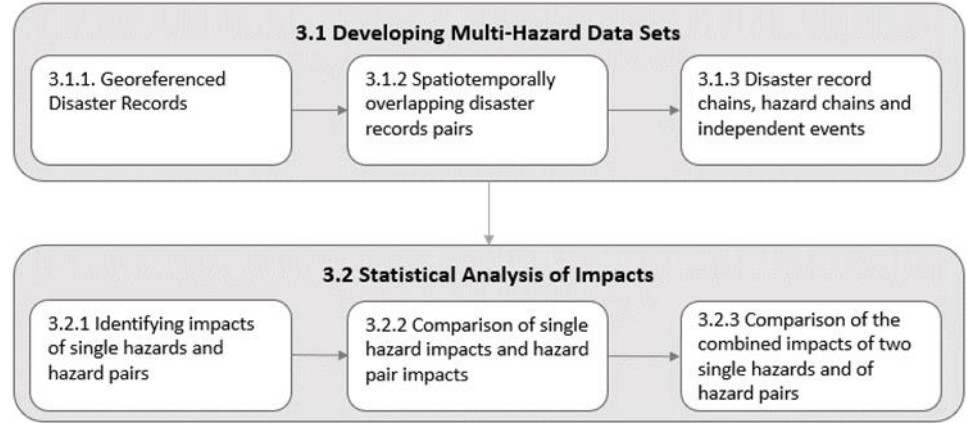

**Figure 1 The two main parts and sub-steps of the methodology**

### 3.1 Developing Multi-Hazard Data Sets

In this section, we describe the three steps to identify multi-hazard events in EM-DAT. The python code of the algorithm can be found on GitHub (link to be added upon publication).

### 3.1.1 Georeferenced disaster records

In the first step, we create a data set of georeferenced disaster records. The GDIS geometries can be linked to the EM-DAT data set via the unique disaster event identifier that is present in both data sets. Given the properties of, and guidelines for,





EM-DAT and GDIS, we include disaster records that fall within the period 2000 – 2018 and belong to one of the seven

disaster types listed in the second column of Table 2. The associated disasters do not follow the main classification system of

EM-DAT but appear to correspond to either the disaster type or the disaster subtype. For consistency we map them to a

disaster type. If the associated disasters cannot be mapped to one of the seven disaster types we focus on in this study, we

exclude the record from the analysis. In the rest of the paper we use the term hazard types instead of disaster type to be in

line with terminology of the disaster risk field (PreventionWeb 2023). We use nine different hazard types that capture

different combinations of disaster type and subtype as well as the associated disasters. We use the same terms for the hazard

types as in a previous paper (Claassen et al. 2023); they are given in the first column of Table 2.

### 3.1.2 Spatiotemporally overlapping disaster record pairs

In the second step, we use the spatial geometries and dates associated with each disaster record to identify spatiotemporal

overlaps. We explain the algorithm with the illustration in Figure 2. This example has five disaster records A – E. Figure 2a

shows the relevant information for the algorithm as stored in EM-DAT. Each disaster record has a start date as well as a

main hazard type and optionally one or two associated hazard types.  The algorithm works as follows:

1. We create a list of all possible pairwise combinations of disaster records per country. Suppose disaster records A – E are in one country, then all possible combinations would be "A, B", "A, C", "A, D", "A, E", "B, C", "B, D", "B, E" and "D, E".

2. We then assess the spatial overlap for each of these pairs. To this end, we calculate the intersecting area between the disaster records from the spatial geometries, as well as the fractions that the intersecting area constitutes to the total area of each of the individual events. We refer to the higher value of the two as the intersection percentage and use a minimum value as criterion to define spatially overlapping disaster records. Because disasters are unlikely to affect the entire area of an administrative region, the spatial geometries of the disaster records are relatively crude approximations of the impact zones. We reason that the smaller the intersecting area of two footprints, the less likely that the actual disaster impact zones overlap. The idea behind the threshold is to keep only those combinations that have a reasonable likelihood of actually having overlapping disaster zones. We use a threshold of 50% and perform a sensitivity analysis (0%, 25%, 50%, 75%, 100%)[1] on this choice. Given the spatial geometries in Figure 2b, the spatially overlapping pairs would be "A, B", "A, C" and "B, D".

3. We also assess temporal overlap for each overlapping pair from step 1. Because end dates are often missing in the disaster records, we calculate the time difference between the start dates of the pair. We use a maximum time lag as a criterion to define temporally overlapping disaster records. We use a time lag of 3 months and perform a sensitivity analysis on this choice (0 months, 1 month, 3 months, 6 months, 12 months)[2]. Figure 2c depicts the time lags. Suppose all the times between events ($\Delta t_{21}$, $\Delta t_{32}$, $\Delta t_{43}$, $\Delta t_{54}$) are 1 month, the temporally overlapping pairs using a 3 month time-lag would be "A, B", "A, C", "A, D", "B, C", "B, D", "B, E" and "D, E".

---

[1] The criterion is ≥ for all spatial overlap values, except for the 0% value. In this case the criterion is >.

[2] The criterion is ≤ for all time lag values.





4.    We identify all spatiotemporally overlapping disaster record pairs based on the previous assessments of spatial and temporal overlap. In the example, these are "A, B", "A, C" and "B, D".

### 3.1.3 Disaster record chains, hazard chains and (multi-)hazard events

In the third step, we build a data set of disaster record chains and dissolve them into hazard chains. We start by building a disaster record and a hazard chain for each disaster record in EM-DAT. The chains contain all disaster records and hazards
that have potentially contributed to the reported impact through direct or indirect spatiotemporal overlaps. To develop the disaster record chains, we use an iterative algorithm that utilizes the previously identified overlapping disaster record pairs. We explain the algorithm using the previously introduced example from Figure 2a-c:

- For each disaster record (from EM-DAT), we find all pairs of spatiotemporally overlapping disaster records that include this disaster record. In each of those pairs, if the other disaster record is preceding the record of
interest in time, it is considered to be a contributing disaster record. For example, consider D to be the disaster record of interest. Then, B is a contributing disaster record, as the latter occurs earlier in time.

- If the contributing disaster record has in turn another contributing disaster record, we add that one as well, thus considering indirect contributions. Here, A is contributing disaster record to B. Hence, we add A as contributing disaster record to D as well. Adding indirectly contributing disaster records is a recursive process.
For this example the recursive process stops here, because A has no further contributing disaster records.

- The entire disaster record chain then consists of the disaster record of interest and all contributing disaster records ordered in time (A, B, D).

- We obtain the hazard chain by replacing each disaster record by the hazard or set of hazards that it contains (A1, B1, D1, D2).

Figure 2d presents the data set of disaster record chains and hazards chains we obtain from the algorithm for this example.




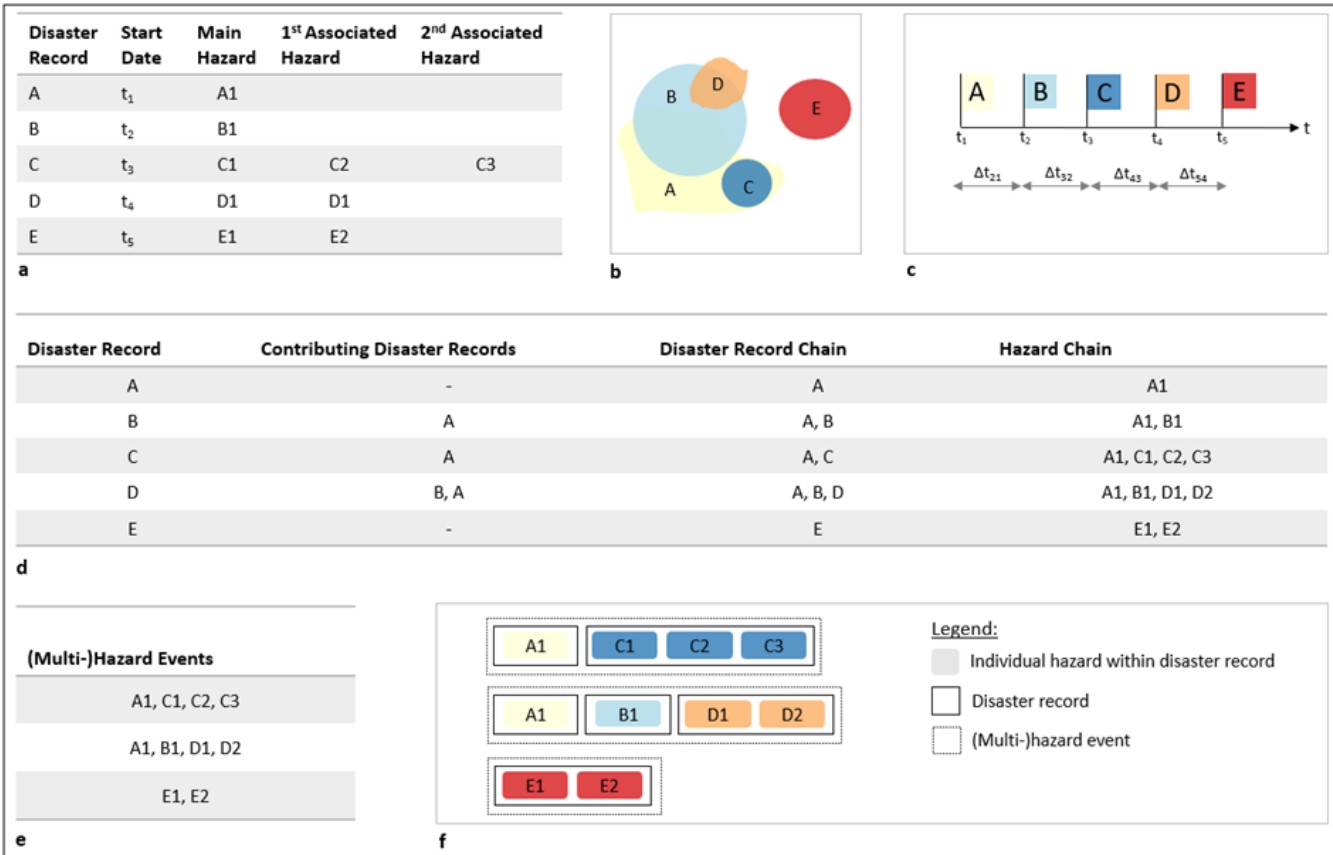

**Figure 2 Illustration of five disaster records  A, B, C, D and E. a) Illustration of relevant fields in EM-DAT; b) Spatial geometries of disaster records; c) Times between disaster records; d) Data set of contributing disaster records, disaster record chains and hazard chains; e) Data set of multi-hazard events; f) Illustration of how individual hazards, disaster records and (multi-)hazard events relate to each other.**

In the last step, we build a data set of (multi-)hazard events. These are those hazard chains, that are not included in another hazard chain. For example, "A1, B1" is fully included in "A1, B1, D1, D2". Therefore, we do not consider "A1, B1" to be an event by itself, but part of the event "A1, B1, D1, D2". Figure 2e shows the resulting events for the example and Figure 2f illustrates how the terms hazard, disaster record and event relate to each other.

### 3.2 Statistical Analysis of Impacts

In this section, we describe the three steps of the statistical analysis.

#### 3.2.1 Impacts of single hazards and hazard pairs

To start with, we create a data set of the human and socioeconomic impacts of single hazards and of hazard pairs. We focus on damages, number of people affected, and number of deaths. For simplicity, we use the term impact to refer to them





collectively. We create the data set by selecting all hazard chains consisting of one or two hazards. These would be "A1", "A1, B1" and "E1, E2" in the example of Figure 2d. If the chain consists of one hazard, we record a single hazard impact. If a chain consists of two hazards, we record a hazard pair impact. Note that a chain of hazards can belong to one disaster

record or two disaster records. If they belong to two disaster records, we sum the impacts to obtain the overall impact of both hazards.

### 3.2.2 Comparison of single hazard impacts and hazard pair impacts

In the second step, we use boxplots to examine the distributions of impacts of single hazards and the impacts of hazard pairs for different hazard types. Then, to assess whether impacts are different for single hazards and for hazard pairs, we compare

the confidence intervals (CIs) of their means. We construct the CIs with a percentile bootstrap (N=10,000). If the CIs overlap, we conclude that the difference in impacts is not statistically significant. If there is no overlap, the difference is statistically significant.

### 3.2.3 Comparison of the combined impacts of two single hazards and of hazard pairs

In the third step, we do not examine data distributions with boxplots, because we do not have a direct data set of combined

impacts of two single hazards. However, we use the same CI approach as above to compare whether differences in impacts are statistically significant or not. Here, the mean combined impacts of two single hazard types is given by the sum of the mean impacts of two single hazard types.

## 4. Results

First we show the main results related to the prevalence of multi-hazards and their share of impacts when considering both

the information of associated hazards in EM-DAT as well as spatiotemporal overlaps between the disaster records. These are the results of method step 3.2.3. For more details on intermediate results of method steps 3.1.2 and 3.1.2, we refer to sections A.1 and A.2 of the appendix. Thereafter, we show the results of the statistical analysis of impacts which were described in method steps 3.2.1 - 3.2.3.

### 4.1 Prevalence of Multi-Hazards and their Share of Impacts

The subset of EM-DAT that we are analysing contains 5868 disaster records, of which 74% have one hazard, 22% have two hazards, and 4% have three hazards. This corresponds to 74% single hazards and 26% multi-hazards when following the approach of Lee at al. (2024) to reclassifying EM-DAT records. Here, we describe how these numbers differ when considering spatiotemporal overlaps between disaster records and allowing multi-hazard events to consist of multiple disaster records.



A challenge arises when accounting for spatiotemporal overlaps between records: individual hazards can often no longer be uniquely assigned to one event but are part of multiple events. This is different to the case when each disaster record constitutes an event and individual hazards are confined to this event. Therefore, we shift the perspective from the level of disaster record to level of individual hazard. We exemplify this with numbers. The data subset contains a total number of

7605 hazards (Using the values from the first paragraph: 74% * 5868 disaster records * 1 hazard + 22% * 5868 disaster records * 2 hazards + 4% * 5868 disaster records * 3 hazards = 7605 hazards). 57% of the 7605 hazards are single-hazards and 43% are part of multi-hazards according to Lee's classification approach. These multi-hazards caused 57% of the total damages, 40% of the total people affected and 49% of the total deaths globally, which is comparable to Lee et al.'s results who assessed all natural hazards in EM-DAT from 1900 - 2023.


We find that within our data set the proportion of hazards that are part of multi-hazards is likely higher than the 43% detected using Lee et al. (2024) due to spatiotemporal overlaps of the disaster records in EM-DAT. However, there is uncertainty related to the choice of overlap criteria Figure 3a shows the proportion of hazards that are part of multi-hazards for different assumptions for spatiotemporal overlap. The lower the criterion for minimum spatial overlap and the higher the

criterion for maximum time lag, the higher the proportion of hazards that are part of multi-hazards. For example, the proportion of hazards that are part of multi-hazards increases to 61% when assuming a spatial overlap of at least 50% and a time lag of at most 90 days. Together they cause 78% of the total damages, 83% of the total people affected and 69% of the total deaths globally. When increasing the time lag to 1 year, the proportion increases to 76% and together they cause 91% of the total damages, 91% of the total people affected and 95% of the total deaths globally.


Similarly, the number of events that we identify depends on the choice of spatiotemporal overlap criteria. The data sets of identified (multi-)hazard events for different criteria can be found at Zenodo (link to be added upon publication). Here, we present the results for a spatial overlap of at least 50% and a time lag of at most 90 days as Figure 3 gives an indication of sensitivity. In this case, we identify 2291 (multi-)hazard events of which 65% are single-hazard events and 35% are multi-

hazard events. It may seem contradictory that the number of multi-hazard events is lower than the number of EM-DAT disaster records, whereas the number of individual hazards that are part of multi-hazards is higher. This is because a disaster record contains at most three hazards, whereas the multi-hazard events that we derive can have more hazards. Overall, we find 218 event types with different hazard combinations, which consist of up to 32 individual hazards from 5 different hazard types.



**Figure 3 Share of (a) hazards that are part of multi-hazards when accounting for spatiotemporal overlaps between the disaster records in EM-DAT for different values of minimum spatial overlap and maximum time lag as well as their (b) total damages, (c) total people affected and (d) total deaths.**

## 4.2 Comparison of Single Hazard and Hazard Pair Impacts

Now we present the statistical analysis of impacts. Again, we present the results for a spatial overlap of at least 50% and a time lag of at most 90 days. Figure 4 shows the boxplot distributions of impacts of single hazards and hazard pairs for different hazard types as well as the mean values with 95% confidence interval (CI). We only show impact types and hazard types with sample sizes $N \geq 50$ in an attempt to capture the broad range of underlying hazard intensity, exposure and vulnerability conditions in which the impacts arise. There are eight combinations of impact type and hazard pair type that fulfil this criterion. For extreme winds and floods, sufficient data are available for total damages, total deaths and total number of people affected (first column of Figure 4). For floods and landslides as well as consecutive floods, sufficient data are available for total deaths and total number of people affected (second and third column of Figure 4). For earthquake and landslides, sufficient data are available for total deaths (fourth column of Figure 4). Sample sizes are reported in Table C1 and Table C2 in the Appendix.



For all variables (impacts of single hazards and of hazard pairs), the mean value is higher than the 75%-quantile; for 9 of the 16 variables the mean value is even higher than the 95%-quantile. Thus, the majority of impacts are clustered towards the lower end of the impact range while a few very high data points pull the mean upwards. Not surprisingly then, the uncertainties about the mean are large compared to the spread of the distributions: For 10 of the 16 variables, the uncertainty about the mean is larger than the 75% inter-quantile range.

Table 3 lists the results of the comparison of the reported average impacts of hazard pairs with those of the corresponding single hazards. In three cases, the impacts of the hazard pair are significantly higher than those of both single hazards (damages for extreme winds and floods, number of deaths for consecutive floods, and number of people affected for consecutive floods). Also, in three cases, the impacts of the hazard pair are significantly higher than those of one of the single hazards, but not of the other (number of deaths for floods and landslides, number of people affected for floods and landslides, and number of people affected for earthquakes and landslides). Finally, in two cases, the average impacts of the hazard pair are not significantly different than those of either of the single hazards (number of deaths for extreme winds and floods and number of people affected for extreme winds and floods). In no case is the average impact of a hazard pair significantly lower than those of either or both single hazards.

Table 4 lists the results of the comparison of the reported average impacts of hazard pairs with the combined reported average impacts of the two corresponding single hazards. In two cases, the impacts of the hazard pair are significantly higher than the combined impacts of the two single hazards (number of deaths of consecutive floods and number of people affected of consecutive floods). In all other cases, no statistical difference is detected. In no case is the average impact of a hazard pair significantly lower than those of the combined impacts of the two single hazards.




**Figure 4 Boxplots of impact data for single hazards as well as hazard pairs (,) for different impact types and hazard types as well as mean values and their bootstrap 95% CI for single hazards, hazard pairs (,) and the combined impact of two single hazards (+). The rows show different impact metrics. The columns show different hazard types (ew – extreme wind, fl – flood, ls – landslide, eq – earthquake). Only combinations of impact and hazard type with N > 50 are shown.**





**Table 3 Statistically significant differences in average impacts of hazard pairs compared to average impacts of underlying single hazards (ew – extreme wind, fl – flood, ls – landslide, eq – earthquake). A '+' denotes that the value of the hazard pair is higher than the value of the single hazard and a '=' denotes no difference.**

| | Ew, Fl | | Fl, Fl | Fl, Ls | | Eq, Ls | |
|---|---|---|---|---|---|---|---|
| | Ew | Fl | Fl | Fl | Ls | Eq | Ls |
| **Damages** | + | + | | | | | |
| **Number of Deaths** | = | = | + | + | = | | |
| **Number of People Affected** | = | = | + | = | + | = | + |

**Table 4 Statistically significant differences in average impacts of hazard pairs compared to combined average impacts of the two underlying single hazards (ew – extreme wind, fl – flood, ls – landslide, eq – earthquake). A '+' denotes that the value of the hazard pair is higher than the combined value of the two underlying single hazards and a '=' denotes that no difference.**

| | Ew, Fl | Fl, Fl | Fl, Ls | Eq, Ls |
|---|---|---|---|---|
| | Ew + Fl | Fl + Fl | Fl + Ls | Eq + Ls |
| **Damages** | = | | | |
| **Number of Deaths** | = | + | = | |
| **Number of People Affected** | = | + | = | = |

## 5 Discussion

The aim of this study was two-fold. On one hand, we aimed to increase our understanding of the prevalence of multi-hazards in global disasters and their impacts. On the other hand, we aimed to shed light on the potential and limitations of using EM-DAT for multi-hazard analyses, as it is one of the most commonly used impact data bases in disaster risk science. EM-DAT has some multi-hazard information as it reports up to three different hazards per disaster record. However, many well-known multi-hazards are not captured as such but reported as independent events, such as the Guatemala 2010 volcanic eruption and tropical cyclone (Gill and Malamud 2014). In this study, we developed an algorithm to identify multi-hazards, which takes into account the existing multi-hazard information in EM-DAT as well as spatiotemporal overlaps between the disaster records. We obtained a new data set of multi-hazards and their impacts. Furthermore, we analysed and compared impacts of single hazards and hazard pairs using descriptive statistics. Here, we discuss our main findings.

To start with, when we account for spatiotemporal overlaps our analysis suggests that there are up to twice as many hazards that are part of multi-hazards than reported by EM-DAT (Section 4.1). This is the case for a time lag of 1 year and spatial overlap 25%. In addition, multi-hazard events may span multiple disaster records and include up to 32 different hazards, while EM-DAT reports only up to 3 different hazards for each disaster record. However, there is substantial uncertainty related to the identification of multi-hazards. To begin with, the spatial footprints are coarse. Hazards may be affecting the



same administrative area, but not actually be overlapping. In some cases, GDIS resolves hazard footprints at high spatial level, in some cases only at country level. This means we do not know if hazards actually overlapped (see Appendix A.2 for

examples and a discussion). In addition, all individual hazards within a disaster record are associated with the same footprint, even though their footprints may have very different extents. Similarly, the temporal information in EM-DAT is crude and limited. As end dates are partially missing, we used start dates, which is a crude approximation of the actual time lag between hazards. Again, temporal information is provided on disaster record level, but not on the level of individual hazards. Finally, we still lack understanding on how much time lag and overlap should be considered. de Ruiter et al. (2020) suggest

that hazards should be analysed together if direct impacts of a subsequent hazard spatially overlap before recovery from a previous hazard is considered to be completed, but information on recovery times is limited. Despite these uncertainties in the method, multi-hazards are likely underreported in EM-DAT.

Furthermore, the reported multi-hazards contribute a disproportionately high share of total impacts globally compared to

405 single hazards (Lee et al. (2024) and Section 4.1). The question is whether this difference is statistically significant and there are indeed risk dynamics causing disproportionate impact amplifications in multi-hazard events. To answer this, we performed a statistical analysis comparing impacts of hazard pairs with impacts of single hazards as well as combined impacts of two single hazards of the same hazard types. The results differ per combination of impact type and hazard types. However, they are difficult to interpret because of the large uncertainties encountered as well as questionable data quality

owing to known biases and limitations of EM-DAT. In cases where there is no significant difference found between two impact variables, this could either mean that there is indeed no difference between those variables or that there is a difference, but not sufficient evidence in the data set to detect that, for example due to the large uncertainties and right skewed distributions. On the other hand, in cases where there is a significant difference, this could indeed point to an actual difference in impacts, but it is also possibly caused by biases such as systematic double counting of consecutive disasters or

geographical biases. Nonetheless, there are commonalities across all cases: In all cases, the average impact of a hazard pair is as high or higher than the average impact of a single hazard, while the opposite was not found in any cases. Also in all cases, the average impact of a hazard pair is as high as or higher than the combined average impacts of the two underlying single hazards, while the opposite was not found in any cases. This suggests that multi-hazard interactions leading to increased impact tend to outweigh multi-hazard interactions leading to decreased impact.

The results of the statistical analysis suggest that there are different patterns of impact-generating mechanisms in multi-hazard events for different types of impact and hazard types. We observed four different patterns which we call archetypes, inspired by the field of system dynamics which uses the term to describe certain common dynamics that seem to recur in many different settings (Senge 1990). We provide short descriptions of each archetype and hypothesize possible

explanations, while noting that there is large uncertainty on whether each case falls into a particular archetype.



Archetype 1 - "The whole equals the sum of its parts": The impact of the pair is significantly higher than the impact of both individual hazards, but not significantly different than the combined impact of the two single hazards. Such a pattern would emerge when both hazards in the pair contribute significantly to the impact, but do not significantly affect each other's impact, that is, there is no disproportionately heightened impact by the combined action of the two hazards. This could, for example, be the case for damages of extreme wind and flood pairs which have different damage causing mechanisms to the built environment. Floods tend to affect the interior of buildings and the lower floors, whereas extreme winds tend to damage the exterior of buildings and, in particular, the roof (Amini and Memari 2020).

Archetype 2 - "The whole is greater than the sum of its parts": The impact of the pair is significantly higher than the impact of both individual hazards and also significantly higher than the combined impact of the two single hazards. Such a pattern could arise when both hazards in the pair contribute significantly to the impact and even exacerbate each other's impacts when co-occurring simultaneously or consecutively. This could, for example, be the case when a previous flood increases vulnerability leading to more impacts in a second flood (de Ruiter et al. 2020) or when a previous flood intensifies a second flood due to already saturated soils and thus leading to higher impacts (Berghuijs et al. 2019).

Archetype 3 - "Total loss and damage is already reached by one hazard": The impact of the pair is not significantly different from the impact of either of the single hazards and not significantly different from the combined impacts of two single hazards. Such a pattern could emerge when a hazard causes an ultimate impact to an exposed element, such as total loss for a building or death for a person, or when the impact metric only reports that an element has been affected but not to what degree. In both cases, a second hazard acting on the same elements cannot increase the value of the impact metric anymore. This could potentially be the case for the total number of people affected by extreme wind and flood pairs when the same area is hit by both hazards.

Archetype 4 - "One of the hazards dominates the impact": The impact of the pair is significantly higher than the impact of one hazard but not the other, and not significantly different from the combined impacts of two single hazards. Such a pattern could arise when one hazard is so impactful that, in comparison, the contribution of other hazard is negligible, possibly combined with a "total loss and damage is already reached"-effect. This could, for example, be the case for the number of people affected by flood – landslide pairs and earthquake – landslide pairs. Floods and earthquakes usually occur on larger spatial scales than landslides and trigger landslides within the already affected area so that the landslide will not add to the number of affected people.

There are many limitations preventing a more detailed analysis of multi-hazard impacts. To begin with, we encountered a number of reporting errors or inconsistencies in EM-DAT. In several cases, associated hazards are not reported. For example, the 7.2-magnitude earthquake in Haiti in August 2021 (reported under disaster number 2021-0511-HTI) is followed by a tropical cyclone (Daniels 2021), which is not reported in EM-DAT. Also, in several cases, two consecutive




disaster records report the exact same number of impacts suggesting potential double counting when adding them up. For example, two consecutive earthquakes in Iceland have reported the exact same number of total damages (reported under disaster numbers 00-0076-ISL and 2000-0335-ISL). The fact that we came across these inconsistencies by chance, suggests
that there are many more.

Moreover, the uncertainties in impacts are large, especially related to the mean impact per hazard (pair) type, making it challenging to arrive at general conclusions. In part this is due to a handful of extremely high impact data points pulling the mean value up from the bulk of data points clustered at the lower end of the distribution and introducing uncertainty. In
addition, the sample sizes of impact data are small per event type, that is, per unique combination of number and types of hazards. On one hand, this is due to the many different event types found. On the other hand, impact data are frequently missing in EM-DAT. While we considered 9 hazard types, we could only analyse impacts for four hazard types and four hazard pair types when requiring a sample size of at least N=50. Especially, impacts for extreme temperatures and droughts are missing. For these types, the complexity and difficulty to assess impacts is well known (Wilhite, Svoboda, and Hayes
2007). Finally, the hazards occurred under diverse conditions in terms of hazard intensity, exposure and vulnerability, which can cause a wide range of impacts. Ideally, these factors would be controlled for in the analysis, as done for example by Budimir, Atkinson, and Lewis (2014) in a comparison of fatalities of earthquake-and-landslide hazards as opposed to earthquake single hazards. However, EM-DAT includes insufficient information to do so.

## 6 Conclusion and Recommendations

For this study, we asked "What can we learn about multi-hazards and their risk dynamics from global disaster records?". To answer the question, we developed an algorithm that identifies multi-hazard events using the existing multi-hazard information in EM-DAT as well as spatiotemporal overlaps between the disaster records based on the spatial geometries provided in GDIS. We also conducted a statistical analysis to compare the impacts of hazard pairs with the impacts of (combinations of) single hazards for different impact metrics and hazard types. Despite the well-known limitations of EM-
DAT related to completeness of the records as well as reliability of the impact data, which prevents detailed analyses of the data, we found the database to be useful for exploring high-level patterns at the global scale.

Our approach for identifying multi-hazard events indicated that up to twice as many individual hazards have not occurred as isolated single-hazards, but have been part of multi-hazard events, than it appears from EM-DAT alone. The exact number
remains uncertain, because the information on the spatial and temporal extent of the hazards is coarse and additional hazards may have occurred but not have been reported, either due to reporting biases or because they did not cause a disaster according to the EM-DAT inclusion criteria. Future research should be directed at improving the completeness of EM-DAT as well as at developing high resolution spatial and temporal footprints of the corresponding individual hazards to enable the assessment of spatiotemporal overlaps. While there is uncertainty on whether we correctly identified each hazard as either





being a part of or as not being part of a multi-hazard event, the resulting event sets provide promising case studies for investigating impact-relevant spatiotemporal distances between hazards and their role in compounding impacts in complex multi-hazard events. The statistical analysis indicates that there are different types of multi-risk dynamics which depend on the impact metric as well as the hazard type. In some cases, the average reported impacts of a hazard pair were comparable to those of a single hazard or even two single hazards combined. In other cases, the average reported impacts of a hazard pair

were larger than those of a single hazard or two single hazards combined.

We propose the development of archetypes to capture the different patterns and make initial suggestions for the hazard types analysed in this study. Such archetypes could help decide on the level of complexity to take into account in risk assessments and risk management for a region of interest if relevant hazard types and impact metrics are known. For some types of

hazard and impact, modelling the impact of one dominant hazard may yield a reasonable approximation of multi-hazard impact, while in other cases modelling single-hazards impacts separately and adding them up may yield a reasonable approximation, while yet in other cases, it may be important to take into account interaction effects leading to either increased or decreased impacts compared to a simple sum of individual impacts. However, future research using more reliable data sources is needed to confirm these archetypes, validate them for use in forward-looking risk assessments,

explore potentially additional forms of compounding impacts and expand them for additional hazard types and impact metrics.

Finally, future research should be directed at improving and supporting the information in EM-DAT. To start with, a quality control of the impact data that solely focuses on the most disastrous records could already improve overall reliability because

these records dominate any statistical data analysis that is based on mean values or total values. Another key area to improve the usability of EM-DAT would be to develop data sets of (multi-)hazard intensities as well as exposure and vulnerability that can readily be linked to the disaster records. This could enable a deeper analysis of multi-hazard risk dynamics as factors determining the context in which the hazards occur can be controlled for. Also, impact data sets that are becoming available with novel methods could be readily linkable to EM-DAT. They could be used to enable cross-validation of impacts of

different data sets, and increase the sample size of impacts, which could enable an analysis of impacts for (multi-)hazard types that had to be excluded from the statistical analysis in this study.

**Appendices**

**Appendix A An Exploratory Data Analysis of the Joint EM-DAT and GDIS Data Set**

This section relates to method step 3.1.1. The data set of geo-referenced disaster events covers the period 2000 – 2018 and

the nine hazard types listed in Table 2. It contains 5,868 disaster records. Table A1 shows the availability of data in the optional fields in these records. In case of associated disasters, we assume that empty fields means that no other hazards have taken place. In all other cases, we assume that data are missing. The temporal information is most complete. All events have





a start year and start month as well as an end year. All events other than droughts also have an end month. The exact day is
missing more frequently.

**Table A1 Count of events and the data availability of key variables in the geo-refenced data set**

|     | Event Count | Start Date | End Date | Total Damages | Total Number of Deaths | Total Number of People Affected | Geospatial Footprint |
|-----|-------------|------------|----------|---------------|------------------------|--------------------------------|----------------------|
| fl  | 2782        | 100%       | 92%      | 30%           | 72%                    | 88%                            | 91%                  |
| ew  | 1629        | 100%       | 97%      | 52%           | 74%                    | 73%                            | 77%                  |
| cw  | 198         | 100%       | 47%      | 8%            | 83%                    | 29%                            | 85%                  |
| dr  | 188         | 96%        | 4%       | 40%           | 4%                     | 60%                            | 80%                  |
| hw  | 118         | 100%       | 48%      | 13%           | 84%                    | 31%                            | 79%                  |
| ls  | 353         | 100%       | 98%      | 12%           | 96%                    | 64%                            | 91%                  |
| eq  | 480         | 100%       | 100%     | 42%           | 64%                    | 97%                            | 96%                  |
| vo  | 93          | 100%       | 90%      | 15%           | 13%                    | 89%                            | 82%                  |
| ts  | 27          | 100%       | 100%     | 70%           | 100%                   | 85%                            | 93%                  |

The availability of impact data depends on the hazard type as well as impact type and ranges from 4% for *Total Deaths* due
to droughts to 100% for *Total Deaths* due to tsunamis (Table A1). Human impact is available more than total damages.
Availability also fluctuates across the years and continents. For total deaths it ranges from 64% in 2004 to 78% in 2007 and

from 40% in Oceania to 79% in Asia. For total affected it ranges from 73% in 2010 and 2018 to 87% in 2017 and from 55%
in Europe to 90% in Africa. For damage is ranges from 22% in 2006 to 48% in 2013, and from 14% in Africa to 44% in
Oceania.

The availability of spatial footprints also differs per hazard type, year and continent, but less so than the impact variables. It

ranges from 77% for extreme wind to 96% of earthquakes, from 71% in 2018 to 93% in 2006 and from 77% in Europe to
89% in the Americas. Overall, we could associate 87% (5090/5668) of all events with a spatial footprint which is in line with
the 89% reported by the developers of GDIS (Rosvold and Buhaug 2021).

These results suggests that data for impact and geospatial footprint are not missing at random (MAR) in our extracted data

set which poses a risk of bias in the subsequent analysis. This is in line with the findings of Jones et al. (2022) who identified
the year the disaster occurred, income-classification of the affected country and hazard types as significant predictors of
missingness for human and economic impact variables in a formal statistical analysis of the entire EM-DAT data set.

**Appendix B Pairs of Spatiotemporally Overlapping Disaster Records**

This section relates to method step 3.1.2. Out of the 5,868 disaster records, 5,090 events have spatial footprints. These can be

grouped into 12,951,505 unique combinations of two events. 107,406 pairs have spatial overlap.





Figure B1 shows a histogram of the intersection percentage. Notable is the high number of event pairs with 0% overlap and with 100% when rounded to 2 decimals. The high number of events 0% overlap is likely caused by rounding errors for events that impacted adjacent administrative areas and these event pairs are considered to not be overlapping. Figure B2a

shows an example. The high number of events with 100% overlap is also likely due to the fact that the resolution is on administrative boundary level: As soon as events are within the same administrative district they fully overlap, whereas only large scale impact events that affected multiple administrative districts can partially overlap. Figure B2b-c show examples with different overlap percentages.

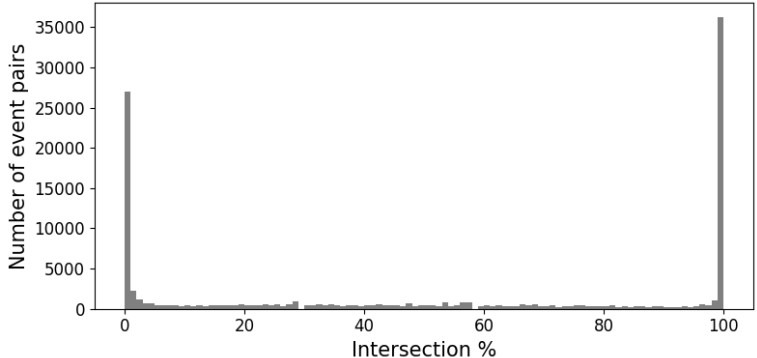


**Figure B1 Histogram of the intersection percentage of the 107406 EM-DAT events with spatial overlap**

There is uncertainty on whether or not the actual impact zones overlap for all pairs of intersecting events, because the spatial footprints are an approximation on the level of administrative regions and the events are unlikely to have affected the entire

region (Rosvold and Buhaug 2021). This uncertainty could potentially reduce by considering the combination of the scale of the natural hazards (e.g., landslides are local events while heat waves and cold waves are regional or national level events), the extent of the damage (e.g., higher damages and fatalities are likely to stem from larger impact zones), and the administrative level of the footprint (e.g., a footprint consisting of multiple district level polygons which have been joint to a greater area is more likely to represent the actual impact area than a footprint consisting of a single country-level polygon).




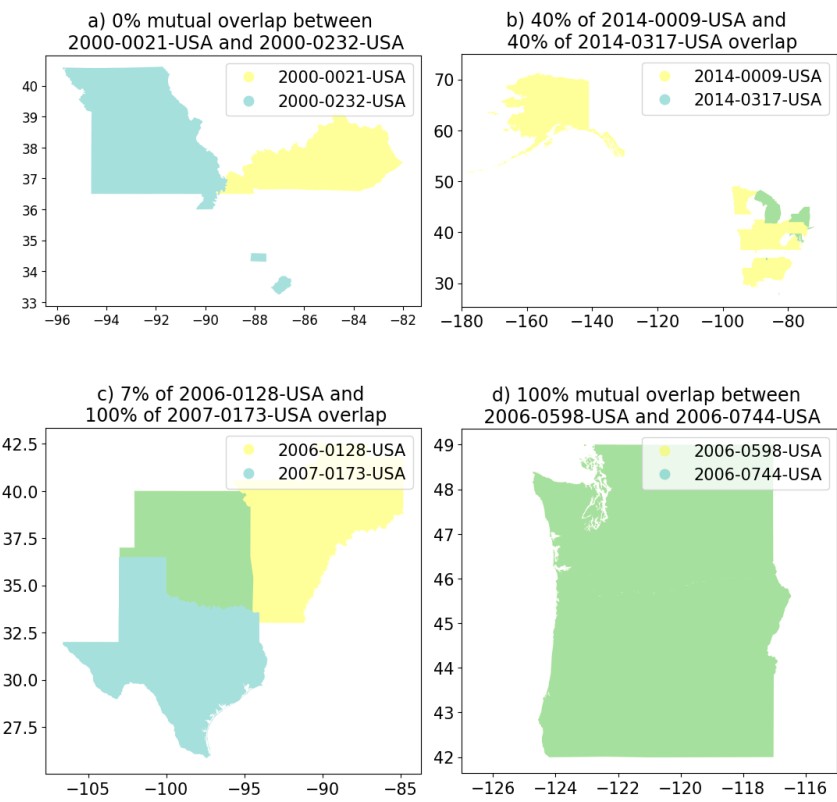

**Figure B2 Example event pairs with spatial overlap. Individual event impact zones are plotted in blue and yellow. The overlapping impact zone is plotted in green.**

For example, in Figure B2b, event 2014-0009-USA is a cold wave and event 2014-0317-USA is a convective storm associated with a cold wave. As these are larger scale weather phenomena the overlapping administrative zone is likely to reflect the actual overlapping impact zone. However, in Figure B2c, event 2006-0128-USA is storm associated with a flood and event 2008-0173-USA is a riverine flood, and, in Figure B2d, event 2006-0598-USA is a riverine flood associated with heavy rain and event 2006-0744-USA is a storm. In these two cases, additional data on impact extent, or by proxy hazard extent, would be required to confirm actual overlap.

Table B1 shows the number of pairs of overlapping events for different spatial and temporal criteria. As expected, the number of pairs of overlapping events is lowest when requiring a high intersection percentage and a low time lag as overlap criteria.

**Table B1 Number of pairs of overlapping events using different spatial and temporal overlap criteria.**

| Time lag \ Intersection | >0% | >=50% | >=100% |
|---|---|---|---|
| **0 days** | 31 | 21 | 17 |
| **1 month** | 1,339 | 758 | 480 |





| | | | |
|---|---|---|---|
| **3 months** | 3,575 | 1,917 | 1,164 |
| **6 months** | 5,865 | 3,023 | 1,798 |
| **12 months** | 11,630 | 6,059 | 3,631 |
| **Any** | 107,406 | 54,712 | 33,245 |

**Appendix C Sample size of impact data for single hazards and hazard pairs**

This section presents sample size for the data used in Sect. 4.2. Table C1 shows the sample sizes for single hazards for a spatial overlap of at least 50% and a time lag of maximum 91 days. Table C2 shows the sample sizes for hazard pairs for a

spatial overlap of at least 50% and a time lag of maximum 91 days.

**Table C1 Sample sizes of impact data for single hazards for a minimum spatial overlap of 50% and a time lag of maximum 91 days (ew – extreme wind, fl – flood, ls – landslide, eq – earthquake, dr – drought, cw – cold wave, hw – heat wave, vo – volcanic activity, ts – tsunami)**

| Hazard | Damages | Number of Deaths | Number of People Affected |
|---|---|---|---|
| fl | 428 | 1102 | 1479 |
| ew | 354 | 527 | 524 |
| eq | 109 | 171 | 281 |
| dr | 52 | 4 | 91 |
| ls | 21 | 230 | 147 |
| cw | 12 | 146 | 50 |
| vo | 10 | 6 | 65 |
| ts | 5 | 6 | 4 |
| hw | 3 | 64 | 23 |

**Table C2 Sample size of impact data for hazard pairs for a minimum spatial overlap of 50% and a maximum time lag 91 days (ew – extreme wind, fl – flood, ls – landslide, eq – earthquake, dr – drought, cw – cold wave, hw – heat wave, vo – volcanic activity, ts – tsunami)**

| Hazard 1 | Hazard 2 | Damages | Number of Deaths | Number of People Affected |
|---|---|---|---|---|
| fl | ls | 178 | 417 | 439 |
| ew | fl | 133 | 205 | 220 |
| eq | ls | 28 | 44 | 56 |
| ew | ew | 16 | 26 | 23 |
| fl | fl | 15 | 67 | 87 |
| ew | ls | 15 | 28 | 31 |
| fl | ew | 12 | 23 | 24 |
| ew | cw | 11 | 22 | 11 |





| | | | | |
|---|---|---|---|---|
| ts | ts | 10 | 15 | 13 |
| hw | dr | 7 | 9 | 2 |
| ls | fl | 6 | 23 | 22 |
| eq | eq | 5 | 8 | 15 |
| dr | hw | 5 | | 3 |
| eq | ts | 4 | 9 | 11 |
| ls | ls | 3 | 12 | 9 |
| eq | fl | 3 | 5 | 5 |
| fl | dr | 2 | 1 | 5 |
| fl | cw | 2 | 5 | 4 |
| dr | fl | 1 | 1 | 6 |
| fl | eq | 1 | 4 | 5 |
| vo | eq | 1 | | 4 |
| fl | ts | 1 | 2 | 3 |
| eq | ew | 1 | 2 | 2 |
| eq | dr | 1 | | 2 |
| ls | eq | 1 | 1 | 1 |
| dr | cw | 1 | 1 | 1 |
| vo | ts | 1 | 1 | 1 |
| dr | eq | 1 | | 1 |
| ew | dr | 1 | | 1 |
| vo | ls | | 1 | 3 |
| hw | ew | | 5 | 2 |
| fl | hw | | 3 | 2 |
| eq | hw | | 1 | 2 |
| hw | fl | | 4 | 1 |
| cw | cw | | 3 | 1 |
| ew | hw | | 3 | 1 |
| vo | fl | | 1 | 1 |
| hw | ls | | 1 | 1 |
| ls | dr | | | 1 |
| cw | eq | | | 1 |
| eq | cw | | | 1 |
| dr | ew | | | 1 |
| vo | vo | | | 1 |
| cw | fl | | 1 | |
| cw | hw | | 1 | |





| ls | ts | | 1 | |
|----|----|--|---|---|
| cw | ls | | 1 | 87 |
| ls | cw | | 1 | 31 |

## Code Availability

The code to develop multi-hazard event data sets as well as to perform the statistical analysis of impacts has been publicly released on GitHub. (Link to be added upon publication).

## Data Availability

The multi-hazard event dataset compiled during the is study, is openly available on Zenodo. (Link to be added upon publication).

## 605 Author Contribution

MCdR and PJW conceived the study. All co-authors contributed to the development and design of the methodology. WSJ analysed the data and prepared the paper, with contributions from all co-authors.

## Competing interests

Some authors are members of the editorial board of NHESS.

## 610 Acknowledgements

WSJ, TT, MCdR and PJW received support from the MYRIAD-EU project, which received funding from the European Union's Horizon 2020 research and innovation programme under grant agreement No 101003276. MCdR also received support from the Netherlands Organisation for Scientific Research (NOW) (VENI; grant no. VI.Veni.222.169). This work used the Dutch national e-infrastructure with the support of the SURF Cooperative using Grant No. EINF-4493.






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
