# Peer review of "What can we learn about multi-hazard impacts from global disaster records?"

_Natural Hazards and Earth System Sciences, 2024_

## Author Comment (AC1)

**RC2 / *AC***

This study makes a valuable and timely contribution to disaster risk science by developing an algorithm that identifies multi-hazard events, utilising information on associated hazards as well as spatiotemporal relationships between disaster records in EM-DAT. The statistical analysis reveals that hazard pairs often lead to greater or at least equal impacts compared to isolated single hazards, although the patterns of impact vary depending on the hazard type and the impact metric. The study proposes developing generic archetypes of multi-hazard risk dynamics to enhance risk analysis and decision-making. While acknowledging the limitations of the EM-DAT database, it demonstrates the database's utility for identifying global patterns of multi-hazard impacts and recommends improvements in data reporting.

The manuscript is generally well-written and addresses an important topic, but several revisions could enhance its clarity, structure, and impact.

*We appreciated the reviewer's time and effort in reading the article and providing helpful comments to strengthen the manuscript before publication.*

I recommend the following adjustments to strengthen the manuscript before publication:

**Methodology and Detail:**

The methodology is sound, but providing more detail about the "statistical methods" previously used (Lines 94–95) would offer readers a clearer understanding of previous research. *Agreed, we will add a sentence on the methods used by Budimir, Atkinson and Lewis (2014).*

Additionally, the manuscript would benefit from justifying the focus on spatial overlap within a single country (Section 3.1.2). For instance, the author could explain why potential transboundary, spatially compounding events, such as those across northern Europe, were not considered (e.g., Fang et al., 2024; De Luca et al., 2017; Berghuijs et al., 2019). *Agreed. We will explain our considerations for making this simplifying assumption.*

The inclusion of Figure 1 is valuable, but expanding its caption to provide more context would help readers understand it without needing to refer back to the main text. *Agreed. RC1 pointed this out as well. We will develop an expanded flow diagram that provides more content.*

**Structure:**

The manuscript would benefit from a more cohesive structure.

For example, moving background information currently placed in the results section (e.g., Line 279) into the methods section would help maintain continuity and allow the

results section to focus more directly on presenting findings. *Does the reviewer mean lines 285 – 287 rather than line 279? We agree that these lines could better be placed in the method section.*

Additionally, keeping the discussion and results sections distinct would improve the paper's flow. Any interpretive content (e.g., Line 301) could be relocated to the discussion. *Agreed. We will move the discussions on sensitivity to the discussion section.*

Furthermore, separating recommendations from the conclusion would also allow the paper to finish on a stronger note, with a distinct conclusion leaving a lasting impression. *Agreed. We will make this change.*

Finally, introducing the archetypes (Line 427 onwards) in more detail in the methods section could help readers appreciate their relevance from the start, enhancing the manuscript's overall coherence. *Thanks for pointing this out. We will introduce the archetypes in the method section already.*

**Writing Style:**

The clarity of the manuscript can be improved by adopting a more concise and direct tone across all sections. For example, removing phrases like "not surprisingly" (e.g., Line 338) and simplifying explanations (e.g., Line 384 regarding spatiotemporal overlaps) would make the writing more focused. Writing all sections more concisely will help maintain a tighter narrative; for example, understanding Lines 480–484 currently requires multiple readings.

Here are three specific examples that would benefit from revision for clarity and conciseness, although consider making changes throughout the manuscript:

- Line 137 – Correct the typographical error "other the other."

- Line 141 – Consider rewriting this sentence to improve its flow.

- Line 488 – Ensure consistent tense usage throughout the text.

*Thanks for pointing this out and providing some examples for clarification. We will re-read the entire manuscript with a focus on concise and direct language and make improvements including the examples mentioned here.*

---

## Author Response (AR1)

**RC1 / AC**

This study presents "an algorithm to identify multi-hazard events which uses the information on associated hazards as well as spatiotemporal relationships between disaster records in EM-DAT." The topic is both relevant and timely, and the manuscript is generally well-written.

We thank the reviewer for taking the time to read the manuscript and for providing constructive feedback that will help us to improve the clarity of the manuscript.

However, I recommend that the authors address the following points before the manuscript is considered for publication:

- The authors assert that there is a lack of understanding regarding historical multi-hazard impacts, yet the introduction does not adequately explain why assessing multi-hazard impacts is particularly challenging. I recommend the authors provide a more detailed explanation of the complexities involved in evaluating multi-hazard impacts. This is a good point. We have included such an explanation and add relevant references in lines 67-74 of the introduction.

- The introduction would benefit from restructuring. After outlining the study's aim, the authors introduce uncertainties associated with the use of EM-DAT, as well as how this global dataset is utilized in the current study. I suggest removing the text from lines 75–99 and incorporating it into sections 2.1 EM-DAT and 5 Discussion for better clarity and flow. Agreed. Uncertainties related to EM-DAT are incorporated in the data section (section 2.1) in lines 130-136 and in the discussion (section 5) in lines 403-409 as well as lines 415-420.

- The criteria for multi-hazard classification in this study are unclear. In section 3.1.2, the authors mention restricting multi-hazard events with an intersecting area smaller than 50%. However, previous studies have outlined various types of multi-hazard events, including preconditioned, triggering, multivariate, spatially compounding, and temporally compounding events. Multi-hazards can also occur in multiple interconnected locations within a limited timeframe. Given this, I am unsure how the authors justify the following statement: "We reason that the smaller the intersecting area of two footprints, the less likely that the actual disaster impact zones overlap. The idea behind the threshold is to keep only those combinations that have a reasonable likelihood of actually having overlapping disaster zones." Thanks for pointing this out. We have included lines 91-94 to clarify the scope of the study.

- Figure 1 requires further elaboration. Including only subsection headings does not sufficiently clarify the overall methodological approach used in the study. I suggest the authors develop a more comprehensive methodological flow

diagram to better explain the proposed approach. Agreed. RC2 pointed this out as well. We have developed an expanded flow diagram (Figure 1) that provides more content. This made us realized that the original steps 3.2.1 and 3.2.2 could better be merged into one step (now step 3.2.2) and that an additional step 3.2.1 would be helpful. We have also included a corresponding section 3.2.1. Step 3.2.3 is new following RC2's suggestions to include the archetypes in the method section.

**RC2 / AC**

This study makes a valuable and timely contribution to disaster risk science by developing an algorithm that identifies multi-hazard events, utilising information on associated hazards as well as spatiotemporal relationships between disaster records in EM-DAT. The statistical analysis reveals that hazard pairs often lead to greater or at least equal impacts compared to isolated single hazards, although the patterns of impact vary depending on the hazard type and the impact metric. The study proposes developing generic archetypes of multi-hazard risk dynamics to enhance risk analysis and decision-making. While acknowledging the limitations of the EM-DAT database, it demonstrates the database's utility for identifying global patterns of multi-hazard impacts and recommends improvements in data reporting.

The manuscript is generally well-written and addresses an important topic, but several revisions could enhance its clarity, structure, and impact.

We appreciated the reviewer's time and effort in reading the article and providing helpful comments to strengthen the manuscript before publication.

I recommend the following adjustments to strengthen the manuscript before publication:

**Methodology and Detail:**

The methodology is sound, but providing more detail about the "statistical methods" previously used (Lines 94–95) would offer readers a clearer understanding of previous research. Agreed.  We will added more information in lines 61-65 on Budimir et al.'s statistical methods (2014).

Additionally, the manuscript would benefit from justifying the focus on spatial overlap within a single country (Section 3.1.2). For instance, the author could explain why potential transboundary, spatially compounding events, such as those across northern Europe, were not considered (e.g., Fang et al., 2024; De Luca et al., 2017; Berghuijs et al., 2019). Agreed. We have added lines 214 – 220 to explain our considerations for making this simplifying assumption.

The inclusion of Figure 1 is valuable, but expanding its caption to provide more context would help readers understand it without needing to refer back to the main text. Agreed.

RC1 pointed this out as well. We have developed an expanded flow diagram (Figure 1) that provides more content. This made us realized that the original steps 3.2.1 and 3.2.2 could better be merged into one step (now step 3.2.2) and that an additional step 3.2.1 would be helpful. We have also included a corresponding section 3.2.1. Step 3.2.3 is new following RC2's suggestions to include the archetypes in the method section.

**Structure:**

The manuscript would benefit from a more cohesive structure.

For example, moving background information currently placed in the results section (e.g., Line 279) into the methods section would help maintain continuity and allow the results section to focus more directly on presenting findings. Does the reviewer mean lines 285 – 287 rather than line 279? These lines have been placed in the data section (lines 152-155).

Additionally, keeping the discussion and results sections distinct would improve the paper's flow. Any interpretive content (e.g., Line 301) could be relocated to the discussion. Agreed. We have moved the discussions on sensitivity to the discussion section (lines 401 and following paragraphs).

Furthermore, separating recommendations from the conclusion would also allow the paper to finish on a stronger note, with a distinct conclusion leaving a lasting impression. Agreed. We have created a separate conclusion and recommendations section.

Finally, introducing the archetypes (Line 427 onwards) in more detail in the methods section could help readers appreciate their relevance from the start, enhancing the manuscript's overall coherence. Thanks for pointing this out. We have introduce the archetypes in the method section (section 3.2.3) and present them in the results (4.3). We have also changed their names to be more coherent among each other and highlighted them explicitly in the abstract.

**Writing Style:**

The clarity of the manuscript can be improved by adopting a more concise and direct tone across all sections. For example, removing phrases like "not surprisingly" (e.g., Line 338) and simplifying explanations (e.g., Line 384 regarding spatiotemporal overlaps) would make the writing more focused. Writing all sections more concisely will help maintain a tighter narrative; for example, understanding Lines 480–484 currently requires multiple readings.

Here are three specific examples that would benefit from revision for clarity and conciseness, although consider making changes throughout the manuscript:

- Line 137 – Correct the typographical error "other the other."

- Line 141 – Consider rewriting this sentence to improve its flow.

- Line 488 – Ensure consistent tense usage throughout the text.

Thanks for pointing this out and providing some examples for clarification. We have re-read the entire manuscript with a focus on concise and direct language. We have streamlined the entire manuscript leading to many track changes throughout. The changes are mainly reordering paragraphs and sentences, removing duplicate information and being more consistent and concise in use of terminology in the text as well as in the headings. This has also led us to include two more sub-figures to Figure 2 in order to keep the different terms and concepts used distinct and clear as well as to merge Tables 3 and 4 into one (now Table 3).

---

## Referee Report (RR1)

This study contributes to disaster risk science by developing an algorithm to identify multi-hazard events using information on associated hazards and spatiotemporal relationships in EM-DAT records. It suggests creating generic archetypes of multi-hazard risk dynamics to enhance risk analysis and decision-making. While acknowledging EM-DAT's limitations, the study highlights the database's value in identifying global multi-hazard impact patterns and recommends improvements in data reporting.

The authors have successfully addressed all concerns raised during the first review, resulting in a manuscript that significantly improves in readability and flow. I recommend the following minor corrections before publication:

- **L190:** The sentence appears unfinished: "disaster types listed in the second column of…"

- **L200:** The sentence appears unfinished: "We use the same terms for the hazard types as (Claassen et al., 2023); they are given in the first column of…"

- **L395:** Add "an" to the sentence: "The aim of this study was to gain (an) understanding of multi-hazards and their compounding impacts by analysing the emergency events database EM-DAT."

- **L488:** Change "than" to "as": "In all archetypes, hazard pairs tend to have at least as much impact  (as) single hazards or combinations of two single hazards, but never less impact."

- **L494:** Separate this long sentence into multiple shorter sentences for improved clarity and grammar. For example: "For some types of hazards and impacts, modeling the impact of one dominant hazard may yield a reasonable approximation of multi-hazard impact. In other cases, modeling single hazard impacts separately and adding them up may also provide a reasonable approximation. However, it may be important to consider interaction effects that could lead to either increased or decreased impacts compared to a simple sum of individual impacts."

- **L507:** Change the sentence to: "In the short term, we recommend improving and supporting the existing information in EM-DAT."

---

## Author Response (AR2)

**Authors response (in green) to referee and editor comments (in black)**

Anonymous referee #2:

This study contributes to disaster risk science by developing an algorithm to identify multi-hazard events using information on associated hazards and spatiotemporal relationships in EM-DAT records. It suggests creating generic archetypes of multi-hazard risk dynamics to enhance risk analysis and decision-making. While acknowledging EM-DAT's limitations, the study highlights the database's value in identifying global multi-hazard impact patterns and recommends improvements in data reporting.

The authors have successfully addressed all concerns raised during the first review, resulting in a manuscript that significantly improves in readability and flow. I recommend the following minor corrections before publication:

We thank the reviewer for taking the time to carefully review the article again.

• L190: The sentence appears unfinished: "disaster types listed in the second column of..." Added "Table 2."

• L200: The sentence appears unfinished: "We use the same terms for the hazard types as (Claassen et al., 2023); they are given in the first column of..." Changed to "as Claassen et al. (2023)

• L395: Add "an" to the sentence: "The aim of this study was to gain (an) understanding of multi-hazards and their compounding impacts by analysing the emergency events database EM-DAT." Done.

• L488: Change "than" to "as": "In all archetypes, hazard pairs tend to have at least as much impact than (as) single hazards or combinations of two single hazards, but never less impact." Done.

• L494: Separate this long sentence into multiple shorter sentences for improved clarity and grammar. For example: "For some types of hazards and impacts, modeling the impact of one dominant hazard may yield a reasonable approximation of multi-hazard impact. In other cases, modeling single hazard impacts separately and adding them up may also provide a reasonable approximation. However, it may be important to consider interaction effects that could lead to either increased or decreased impacts compared to a simple sum of individual impacts." We followed the suggestion closely and broke up the long sentence into these three shorter ones: "For some types of hazard and impact, modelling the impact of a dominant hazard may yield a reasonable approximation of the total multi-hazard impact. In other cases, modelling single-hazard impacts separately and summing them may be sufficient. However, in other situations, considering interaction effects is crucial, as they can either increase or decrease the total impact compared to a simple sum of individual impacts."

• L507: Change the sentence to: "In the short term, we recommend improving and supporting the existing information in EM-DAT." Done.

Editor:

There are a few comments/suggestions I would like to make: We thank the editor for below suggestions to help make the manuscript more clear to readers.

• L.13 p.1: "Overall". I think these values correspond to the 50% spatial and 90 days temporal match. It is not very clear with the current wording. We adjusted to: "We find that 35% of events are multi-hazard events, and 61% of hazards are associated with them, based on a spatial overlap of at least 50% and a time lag of at most three months. The hazards associated with multi-hazard events account for 78% of total damages, 83% of total people affected, and 69% of total deaths."

• L.142 p.5: "when data are not missing at random". I think there is a word missing at the end of the sentence, "locations" maybe? "Missing not at random" is a statistical term and "Data is missing not at random" would be a full sentence. We added the following footnote and reference to explain this term and avoid confusion: "Missing not at random" is a statistical term referring to the likelihood of data being missing being dependent on characteristics of the disaster event (Rubin, 1976), for example, geographic location or disaster type.

• L.279 p.11: "We create the dataset by selecting all events (including the duplicates) consisting of one or two hazards". So what happens to events with more than two hazards? Are they discarded? If, yes, how many events are removed? We added the following paragraph to provide more detail on this: "If an event consists of more than two hazards, we exclude it from this part of the analysis. Such events are either partial duplicates – where the first one or two hazards are already represented by another event – or they correspond to a disaster record with three hazards. For example, we exclude the event "A1, C1, C2, C3". While "A1" is included as single hazard, we cannot include "A1, C1" as a hazard pair because EM-DAT only reports the joint impact of "C1, C2, C3" rather than the individual impacts of "C1", "C2" and "C3" separately. As a result, the disaster record "C1, C2, C3" is not included in the analysis. In total, we had to exclude 1079 (18%) of the EM-DAT disaster records from this part of the analysis due to the aggregated reporting of impacts for all hazards within a record."

• Figure 4 p.15: Please improve the quality of the figure (resolutions). I would also suggest putting all loss metrics on a logarithmic scale to improve readability.

We have increased the resolution of the figure and added a logarithmic scale to one of the plots (number of deaths for extreme winds and floods -panel in the second row, first column). For this case, readability improves indeed. However for the other plots, the readability for the boxplots improves while the readability for the confidence intervals

decreases. As we mainly compare the confidence intervals, we have decided to keep the linear scale for all other plots.

While revisiting this figure we noticed that the plot for number of deaths for floods and landslides was accidently greyed out (missing), although it was being described in the text. We have added this plot.

We also noticed two typo's in table 3: '=' signs which should have been a '<' signs. We corrected these as well as the corresponding sentence parts in the results (l.401), discussion (l. 447), conclusions (l.515) and abstract (l.21).

Finally, we have changed a few more typo's.